# FOXO3a from the Nucleus to the Mitochondria: A Round Trip in Cellular Stress Response

**DOI:** 10.3390/cells8091110

**Published:** 2019-09-19

**Authors:** Candida Fasano, Vittoria Disciglio, Stefania Bertora, Martina Lepore Signorile, Cristiano Simone

**Affiliations:** 1National Institute of Gastroenterology, “S. de Bellis” Research Hospital, 70013 Castellana Grotte (Bari), Italy; fasano.labsimone@gmail.com (C.F.); disciglio.labsimone@gmail.com (V.D.); bertora.labsimone@gmail.com (S.B.); leporesignorile.labsimone@gmail.com (M.L.S.); 2Department of Molecular Medicine, Sapienza University of Rome, 00161 Roma, Italy; 3Division of Medical Genetics, Department of Biomedical Sciences and Human Oncology (DIMO), University of Bari Aldo Moro, 70124 Bari, Italy

**Keywords:** FOXO3a, transcription factors, cellular homeostasis, stress response, nuclear/mitochondrial crosstalk

## Abstract

Cellular stress response is a universal mechanism that ensures the survival or negative selection of cells in challenging conditions. The transcription factor Forkhead box protein O3 (FOXO3a) is a core regulator of cellular homeostasis, stress response, and longevity since it can modulate a variety of stress responses upon nutrient shortage, oxidative stress, hypoxia, heat shock, and DNA damage. FOXO3a activity is regulated by post-translational modifications that drive its shuttling between different cellular compartments, thereby determining its inactivation (cytoplasm) or activation (nucleus and mitochondria). Depending on the stress stimulus and subcellular context, activated FOXO3a can induce specific sets of nuclear genes, including cell cycle inhibitors, pro-apoptotic genes, reactive oxygen species (ROS) scavengers, autophagy effectors, gluconeogenic enzymes, and others. On the other hand, upon glucose restriction, 5′-AMP-activated protein kinase (AMPK) and mitogen activated protein kinase kinase (MEK)/extracellular signal-regulated kinase (ERK) -dependent FOXO3a mitochondrial translocation allows the transcription of oxidative phosphorylation (*OXPHOS*) genes, restoring cellular ATP levels, while in cancer cells, mitochondrial FOXO3a mediates survival upon genotoxic stress induced by chemotherapy. Interestingly, these target genes and their related pathways are diverse and sometimes antagonistic, suggesting that FOXO3a is an adaptable player in the dynamic homeostasis of normal and stressed cells. In this review, we describe the multiple roles of FOXO3a in cellular stress response, with a focus on both its nuclear and mitochondrial functions.

## 1. Introduction

During the evolution of prokaryotic and eukaryotic (unicellular and multicellular) organisms, cells have developed an extraordinary capacity to adapt to adverse changes in their environment. Organisms and their cells are exposed to a broad spectrum of harmful (or stress) events arising from physiological processes and/or external stimuli. Stressor changes and fluctuations in extracellular conditions can lead to damage to the structure and function of macromolecules, namely DNA, RNA, lipids, and proteins. 

All organisms have evolved mechanisms of cellular response to stress, ensuring the survival of stress-adapted cells in challenging conditions and the negative selection of damaged cells [1]. However, while cellular mechanisms of stress response support cellular functions and hence maintain microenvironmental and organismal homeostasis, they can also lead to their impairment over time, thereby promoting aging and/or diseases, such as diabetes, heart disease, neurodegeneration, and cancer [2].

Adaptation to stress is achieved mainly through the induction of highly conserved cellular response processes, which are mediated by stress-activated signaling pathways that define cell fate. These processes involve: i) The metabolic/energetic response system, which is induced by fluctuations in metabolites, including nutrients [3]; ii) the oxidative stress response system, which is activated by excess reactive oxygen species (ROS) and imbalances in the oxidant/antioxidant status within cells [4,5]; iii) the hypoxia response system, which becomes operational in low oxygen conditions [6]; iv) the heat shock response system, which is elicited by exposure to heat and other agents that adversely affect protein folding and impair RNA metabolism [7]; and v) the genotoxic response system, which is triggered by DNA damage [8].

Cellular stress-activated signal transduction networks encompass three main components: sensors that recognize the harmful event; transducers that carry, amplify, and integrate signals; and effectors that adjust cell functions to the perceived signal [9]. Handling the wide range of stresses that cells are exposed to, stress sensors are diverse and highly specialized (e.g., growth factor receptors, cytokine receptors, cyclic adenosine mono-phosphate (cAMP) and cyclic guanosine mono-phosphate (cGMP) receptors, cellular adhesion receptors, ion channels, etc.). Signal transduction often relies on systems based on phosphorylation cascades, providing fast cellular response. Stress transducers include molecules that can be either membrane-bound or soluble and directly regulate downstream intracellular effectors. The main effectors of signal transduction pathways are transcription factors, along with other factors that assist and coordinate gene expression through chromatin remodeling, mRNA synthesis, mRNA stability, and translational regulation [10]. Hence, the response to cellular stress perturbations is mediated by an orchestrated gene expression program leading to DNA repair, cellular senescence, or apoptosis.

The forkhead box (FOX) family of transcription factors includes more than 100 members, which are characterized by the presence of a conserved forkhead DNA-binding domain that enables the regulation of gene transcription. FOX transcription factors can be divided into 19 subclasses based on phylogenetic analysis (FOX A-S) [11]. Class O subfamily (FOXO) proteins are conserved in all metazoans, from worms to humans, which emphasizes their role as key regulators of cell survival [12,13]. 

In particular, invertebrates only have one *FOXO* gene, namely abnormal dauer formation (*DAF-16*) in *Caenorhabditis elegans* and *dFoxO* in *Drosophila melanogaster*. *DAF-16* was first identified in *C. elegans* as a mediator of dauer larval development under environmental cues, suggesting that it has a role in cellular stress response [14]. Further studies revealed that *DAF-16* is required for promoting cell survival and longevity through the regulation of response to several stimuli, including oxidative stress [14].

The role of *FOXOs* in stress response was also confirmed in *D. melanogaster*, where it was shown that *dFoxO* activates several components of the heat shock response family following oxidative stress [15]. 

The results obtained in *C. elegans* and *D. melanogaster* have been extended and validated in the more complex mammalian systems. Mammals have four FOXO proteins, namely FOXO1, 3, 4, and 6, which show high sequence similarity. These factors are ubiquitously expressed, and the expression level of each protein is tissue-specific, hence reflecting potential differences in cellular activity [16]. 

In mice, disruption of both *Foxo3* alleles led to age-dependent reduced fertility, although it has been suggested that functional redundancy between the four *Foxo* genes could compensate for the absence of a stronger phenotype [17]. 

In humans, Forkhead box protein O3 (FOXO3a) activity has been associated with age-related phenotypes and its role has been linked to increased lifespan through the modulation of stress responses upon oxidative stress, DNA damage, nutrient shortage, and caloric restriction. Notably, functional alterations of FOXO3a have been correlated with degenerative diseases, premature aging, and poor prognosis in several types of cancer [18,19,20,21]. The use of different animal models allowed the identification of FOXO3a as a central mediator of cellular response to stress [22]. Indeed, FOXO3a is a core regulator of cellular homeostasis, stress response, and longevity [23], as it can modulate a variety of cellular processes, integrating inputs coming from energy, growth factor, and stress signaling cascades. The main processes regulated by FOXO3a include cell cycle progression [24,25], DNA repair [26], autophagy [27], ROS detoxification [28], and apoptosis [27].

The activity of FOXO3a is controlled by various post-translational modifications (PTMs), which determine its subcellular localization. Current knowledge suggests that FOXO3a PTMs, such as phosphorylation, acetylation, methylation, and ubiquitination, are the common thread of FOXO3a involvement in cellular homeostasis regulation [29,30]. In particular, FOXO3a PTMs drive FOXO3a activity towards the nuclear and/or the mitochondrial compartment in response to stress stimuli. After undergoing activating and inhibiting modifications, FOXO3 induces a specific set of genes involved in the regulation of various cellular processes. In fact, cytoplasmic FOXO3a is inactive and is shuttled either to the nucleus or to the mitochondria to exert its transcriptional functions [29,31].

In this review, we will describe in detail the multiple roles of FOXO3a in cellular stress response, with a focus on the signaling pathways regulating its activity and on the crosstalk between its nuclear and mitochondrial functions. 

## 2. Upstream Regulation of FOXO3a

The crucial role of FOXO3a seems to be the integration of information from multiple upstream signals, enabling an organism to maintain its tissue homeostasis during stress. FOXO3a receives signals from a variety of cellular stimuli (growth factors, metabolic stress, oxidative stress) regulating its activity mainly through reversible PTMs, which include phosphorylation, acetylation, methylation, and ubiquitination. FOXO3a transduces these signals into responses that modulate protein–protein interactions and orchestrate the spatial and temporal expression of genes, resulting in regulation of the cell cycle, cell survival, and metabolism. Reversible FOXO3a PTMs can have synergistic or competitive effects on its stability, subcellular localization, DNA binding, and/or transactivation function (Figure 1) [31].

### 2.1. The PI3K-AKT-FOXO3a and AMPK-FOXO3a Axes in Nutrient Signaling and Energetic Stress

Glucose deprivation induces major metabolic stress signals, which promote a decrease in cellular generation of ATP and other biosynthetic precursors, like nucleic acids and fatty acids, leading to ROS overproduction [32]. The most important sensors playing a guardian role in the regulation of cellular metabolism and mitochondrial homeostasis are the phosphatidyl inositol 3-kinase (PI3K) and the 5′-AMP-activated protein kinase (AMPK) signaling pathways [33].

Activation of the PI3K pathway is triggered by several stimuli, including insulin, growth factors, and erythropoietin. These lead to the activation/phosphorylation of the downstream effector protein kinase B (PKB/AKT), which in turn regulates the activity of different targets, including kinases and transcription factors, such as FOXO3a [27].

Regulation of FOXO3a activity by the PI3K-AKT signaling pathway, which plays a crucial role in glucose metabolism, is evolutionarily conserved [34,35]. Active AKT translocates into the nucleus, where it is responsible for the phosphorylation of FOXO3a at three conserved residues (Thr 32, Ser 253, and Ser 315) [27]. This affects FOXO3a subcellular localization, leading to its association with the 14-3-3 nuclear export protein. In turn, this binding excludes FOXO3a from the nucleus, promoting its cytoplasmic accumulation and subsequent degradation. Consequently, AKT-mediated FOXO3a phosphorylation inhibits FOXO3a transactivation activity [27].

An additional downstream effector of PI3K activity is the serum and glucocorticoid-regulated kinase (SGK), which directly phosphorylates FOXO3a at sites overlapping those recognized by AKT. Both AKT and SGK phosphorylate Thr 32, but SGK exhibits a preference for Ser 315, whereas AKT phosphorylates Ser 253 more efficiently. Ser 315 phosphorylation primes FOXO3a for further phosphorylation by casein kinase 1 (CK1), which is dependent on AKT/SGK phosphorylation and accelerates its cytoplasmic sequestration [36]. Conversely, the phosphatase and tensin homolog deleted on chromosome 10 (PTEN) is one of the main antagonists of the PI3K/AKT pathway [37]. Overall, the PI3K/AKT pathway plays an important role in regulating glycolytic metabolism. 

Another important cell energy sensor and regulator of energy homeostasis is AMPK, which is highly conserved across eukaryotic species. It is activated by energy stress in response to decreased production of ATP (e.g., glucose restriction (GR), hypoxia, oxidative stress) or accelerated ATP consumption (e.g., cell proliferation, anabolism, muscle contraction), which are sensed as changes in the ADP/ATP or AMP/ATP ratios. AMPK is a heterotrimeric complex composed of three subunits: A catalytic subunit (α) and two regulatory subunits (β and γ). Activation of AMPK occurs in response to an increased cellular AMP/ATP ratio, which arises during cellular metabolic stress [38,39,40].

AMPK can restore energy balance by inhibiting ATP-consuming biosynthetic pathways (gluconeogenesis, fatty acid synthesis) and by activating pathways that regenerate ATP through macromolecule breakdown (glycolysis, fatty acid oxidation). Moreover, the ability of AMPK to inhibit protein synthesis is mediated in large part by direct inhibition of mechanistic (or mammalian) target of rapamycin complex 1 (mTORC1) [41,42]. mTOR is a central integrator of nutrient and growth factor signals that activates several biosynthetic pathways and stimulates cell growth [43].

In mammals, the AMPK-FOXO3a axis has a central role in the balance between energy-consuming and energy-producing pathways. Activation of FOXO factors by AMPK promotes preferential expression of a gene expression program that enhances cellular stress resistance [38,44]. AMPK can directly regulate FOXO3a by phosphorylation at different residues (Thr 179, Ser 399, Ser 413, Ser 555, Ser 588, and Ser 626). These PTMs do not affect FOXO3a subcellular localization, indicating that AMPK influences FOXO3a activity only when it is located in the nucleus [38]. FOXO3a phosphorylation on Ser 626 by AMPK enhances its affinity for the transcriptional activator CREB-binding protein (CBP) and its paralog p300 (CBP/p300), which possess histone acetyltransferase activity and acetylate histone tails, resulting in chromatin remodeling, but can also modify non-histone proteins, including transcription factors [45]. CBP/p300 can indeed acetylate FOXO3a, reducing its DNA binding ability and thus its transcriptional activity [46].

Importantly, activation of the mitogen activated protein kinase kinase/extracellular signal-regulated kinase (MEK/ERK) and AMPK pathways leads to FOXO3a phosphorylation on Ser 12 and Ser 30, respectively, inducing FOXO3a translocation into the mitochondria in metabolically stressed cancer cells. This leads to FOXO3a binding to mitochondrial DNA (mtDNA), together with transcription factor A mitochondrial (TFAM), mitochondrial RNA polymerase (mtRNApol), and sirtuin 3 (SIRT3), and, subsequently, to the induction of mitochondrial gene expression, with the final effect of sustaining the healthy and functionally active state of mitochondria [47].

Additionally, AMPK can influence autophagic activity in a transcriptional manner. In fact, under stress conditions, AMPK directly phosphorylates FOXO3a, which regulates genes implicated in the autophagy process [38]. This activity competes with that driven by the mTOR pathway, which phosphorylates other members of the FOX family, namely forkhead box K1 (FOXK1) and forkhead box K2 (FOXK2), under conditions of high nutrients. Upon activation by mTOR, FOXK1 and FOXK2 translocate into the nucleus, where they compete with FOXO3a-binding sites, down-regulating its targets involved in autophagy [48]. As a result, activation of AMPK under starvation conditions leads to FOXO3a phosphorylation and to the up-regulation of autophagy genes, while repression of mTOR induces FOXK exclusion from the nucleus, allowing increased transactivation of FOXO3a target genes.

Moreover, the expression of several autophagy genes (microtubule-associated protein light chain 3B, *LC3B*; GABA type A receptor associated protein like 1, *Gabarapl1*; Bcl2 interacting protein 3, *Bnip3*; and Bcl2 interacting protein 3 like, *Bnip3l*) mediated by Foxo3a can contribute to rhythmic autophagy activation in a clock-dependent manner, confirming the involvement of Foxo3a in circadian induction of autophagy [49]. A recent study in a mouse model also showed a crucial role of Foxo3a in coupling the circadian clock to metabolism. In particular, the authors reported that insulin regulates the molecular clock in a PI3K- and Foxo3a-dependent manner, suggesting a key role of the insulin-Foxo3a-clock signaling pathway in the regulation of circadian rhythms [50].

Altogether, these data highlight the importance of the AMPK-FOXO3a axis in the modulation of crucial cellular processes, including cell metabolism, stress resistance, and autophagy. 

### 2.2. The MAPK-FOXO3a Axis in Oxidative Stress

Oxidative stress is an imbalance between ROS production and cellular antioxidant defenses [51]. ROS comprise O_2_-derived reactive molecules that are normally produced either in physiological conditions (e.g., during aerobic respiration) or after different types of exogenous stress (e.g., exposure to ionizing radiations). In fact, lipid and carbohydrate catabolism (e.g., aerobic glycolysis and fatty acid β-oxidation), which occurs primarily in mitochondria, is the main endogenous source of oxidative stress and ROS [52]. 

The term reactive oxygen species comprises classes of endogenous and exogenous small short-lived and highly reactive signaling molecules. Endogenous classes of ROS include superoxide radicals (O_2_•−), hydrogen peroxide (H_2_O_2_), hydroxyl radicals (•OH), and singlet oxygen (^1^O_2_) [53]. 

In mammals, endogenous ROS are generated from several sources, including the Nox family of nicotinamide adenine dinucleotide phosphate (NADPH) oxidases and the mitochondrial respiratory chain [54,55]. Moreover, generation of endogenous ROS can be achieved by different ROS stress inducers, including hypoxic stress [56], which stimulates ROS production by mitochondria, and metabolic defects, which are generated by metabolic reactions [57]. The maintenance of cellular homeostasis in spite of ROS production is mediated by signaling pathways playing important roles in ROS catabolism and oxidative injury repair [58]. ROS have been implicated in the activation of various cellular signaling pathways and transcription factors, including PI3K/AKT, mitogen-activated protein kinases (MAPK), and the tumor suppressor p53 (p53), which can activate cell survival and/or cell death processes, such as autophagy and apoptosis. Overall, these signaling pathways converge on FOXO3a, modulating its activity in response to high ROS levels. In metabolically stressed cells, glucose deprivation reduces ATP production and leads to ROS overproduction, which in turn modulate AMPK and AKT signaling pathways. AKT, a major player in insulin signaling cascades, can be directly regulated by redox reactions at its cysteine residues Cys 297 and Cys 311, which has an insulin mimetic effect. 

Oxidation of these cysteines induces the formation of intramolecular disulfide bonds that increase AKT affinity for protein phosphatase 2A (PP2A), enhancing AKT dephosphorylation and thus its inactivation [59]. Hence, ROS-induced inactivation of AKT activates the FOXO3a transactivation function. 

The main homeostatic pathway modulating cellular response to oxidative stress is the MAPK-FOXO3a axis. MAPK cascades represent a central signaling pathway that coordinates intracellular communication between membrane receptors and their nuclear or cytoplasmic targets [60]. MAPKs regulate a wide range of cellular homeostatic processes, including proliferation, differentiation, survival, apoptosis, and response to stress conditions (oxidative stress, genotoxic stress, heat shock, growth factor deprivation). MAPKs comprise four distinct kinase groups: i) Extracellular signal-regulated kinase 1 and 2 (ERK1/2); ii) c-Jun N-terminal kinase 1 to 3 (JNK1-3); iii) p38MAPK α, β, γ, and δ (p38α-δ); and iv) ERK5 [61,62]. The ERK1/2 cascade transmits mostly mitogenic signals, whereas p38 and JNK cascades transmit mainly stress signals [63].

In response to oxidative stress, FOXO3a is phosphorylated by several kinases that promote its nuclear accumulation. For example, FOXO3a can be phosphorylated by macrophage stimulating 1 (MST1), an upstream activator of the MAPK pathway. This disrupts the interaction between FOXO3a and 14-3-3, thereby promoting FOXO3a nuclear translocation [30].

Moreover, growth factor treatment or expression of catalytically active upstream MAPKs, both of which activate ERKs, induce ERK-dependent phosphorylation of FOXO3a at Ser 294, Ser 344, and Ser 425, promoting cell proliferation and tumorigenesis. In fact, it has been proposed that ERK-phosphorylated FOXO3a is a target of mouse double minute 2 homolog (MDM2)-mediated ubiquitination, leading to FOXO3a proteasomal degradation and, subsequently, to suppression of genes involved in apoptosis and cell cycle arrest, like Bcl2-like protein 1(*BIM*) and cyclin dependent kinase inhibitor p27 (*p27*) [64]. 

The JNK cascade is one of the MAPK signaling pathways activated by environmental stress (oxidative stress, ionizing radiation, DNA damage). It is involved in the control of a number of cellular processes, including proliferation, embryonic development, and apoptosis. After activation, JNK mediates FOXO3a phosphorylation at Ser 574, promoting FOXO3a nuclear import and modulation of its transcriptional activity. Moreover, JNK can indirectly activate FOXO3a by repressing PI3K-AKT activity, leading to FOXO3a nuclear translocation [12,65]. 

Upon activation by JNK or MST1, FOXO3a can interact with other transcription factors, such as p53 [66] or myelocytomatosis oncogene cellular homolog(c-MYC) [67], and thereby indirectly regulate the expression of their target genes related with cell death pathways. Notably, in response to chemotherapy, another MAPK, namely MAP-kinase p38 (p38), is able to phosphorylate FOXO3a, promoting its nuclear translocation [68].

For these reasons, modulation of signaling pathways involved in mediating FOXO3a cellular response to oxidative stress may represent an important opportunity for the treatment of diseases in whose pathophysiology ROS are crucial players.

### 2.3. The HIF1α-FOXO3a Axis in Hypoxic Stress

Aerobic organisms produce energy through oxygen consumption. For this reason, hypoxic conditions generate significant stress in cells. In addition, oxygen deprivation is also linked to accumulation of free radicals, which cause additional damage to proteins and DNA. Cells respond to hypoxia by activating survival mechanisms, such as cell cycle arrest, energy consumption reduction, and secretion of proangiogenic factors [6]. These processes are regulated by various stress-activated pathways, including heat shock response, mTOR signaling, and gene regulation by hypoxia-inducible factor (HIF) [69]. 

HIF is a heterodimeric transcription factor (formed by an α and a β subunit), considered as the central regulator of oxygen detection and subsequent cellular adaptation [69,70]. At physiological oxygen levels (normoxia), HIF is degraded through a proteasomal-dependent mechanism that requires hydroxylation of its proline residues by HIF prolyl hydroxylases (PHDs) [71]. In conditions of low oxygenation (hypoxia), HIF is stabilized and can transcribe genes with adaptive functions [72]. Additional evidence shows that HIF regulation can also be mediated by mitochondria since it has been demonstrated that ROS generated by these organelles can inhibit its degradation [73]. Moreover, HIF activity can be stimulated by sirtuins [74], a family of stress-responsive histone deacetylases that influence gene transcription, metabolism, DNA repair, and longevity, acting as sensors of the cellular redox state, because they respond to the ratio of oxidized/reduced nicotinamide adenosine dinucleotide (NAD+/NADH) [75].

Several findings suggest that FOXO3a is a crucial target of HIF-mediated sensing of cellular stress, since hypoxia can directly induce FOXO3a expression and nuclear localization, and these processes are associated with HIF activation [76,77]. It has been demonstrated that FOXO3a up-regulation by HIF occurs at the transcriptional level since HIF1α silencing decreases FOXO3a mRNA and protein levels but does not affect its phosphorylation status nor its subcellular localization [77,78]. The hypothesis that FOXO3a is a direct target of HIF is further supported by evidence showing direct protein–protein interaction between FOXO3a and HIF1α and by the presence of three conserved HIF-binding consensus sites—called hypoxia-responsive elements (HREs)—in the *FOXO3* promoter [78,79].

Other studies revealed that FOXO3a activity in response to hypoxia is finely regulated by multiple upstream factors, besides HIF. For example, FOXO3a can be negatively regulated by AKT, whose deregulation following hypoxia decreases FOXO3a phosphorylation, thereby inducing its export into the cytoplasm [77]. FOXO3a can also be a target of PHD hydroxylation, which promotes its proteasomal degradation [80]. Moreover, FOXO3a activity can be modulated by *PTEN*, since it has been demonstrated that loss of this gene decreases FOXO3a nuclear localization as well as its transcriptional activity [81]. 

### 2.4. Upstream Regulation of FOXO3a in Heat Shock Stress 

A sudden variation in external temperature, commonly known as “heat shock”, has deleterious effects on a cell’s internal organization since it can lead to protein unfolding, cytoskeleton disruption, loss of correct organelle localization, and breakdown of intracellular transport processes [82,83,84]. Moreover, at the nuclear level, heat shock can also impair RNA metabolism, subsequently leading to a decrease in protein translation and DNA replication [85,86].

Eukaryotic heat shock response is a highly conserved transcriptional program that relies on the immediate synthesis of an array of cytoprotective factors in the presence of thermal and other environmental stresses (e.g., oxidative stress, exposure to toxic substances) [7]. The vast majority of these factors are molecular chaperones, normally referred to as heat shock proteins (HSPs), that are able to assist the refolding or elimination of denatured proteins. Up-regulation of HSPs after heat shock relies on a specific transcription factor, called heat shock factor 1 (HSF1), which can activate the transcription machinery upon stress. 

Although its role in cellular homeostasis has been mainly linked to the transactivation of genes encoding HSPs, recent genome-wide studies have revealed that HSF1 also regulates cellular response to oxidative stress, hypoxia, and metabolic imbalance [87]. In physiological conditions, HSF1 is kept in its inactive form in the cytoplasm through physical interaction with HSP90, in coordination with HSP70/HSP40 (which can directly bind HSF1 or promote its loading to the HSP90 complex). In stress conditions, the increasing amount of unfolded proteins causes the release of chaperones from HSF1, which then oligomerizes, translocates into the nucleus, and binds to specific heat shock responsive elements (HSEs) on the DNA [88]. In a recent study, a novel HSE-binding site for HSF1 was identified in *FOXO3* intronic single nucleotide polymorphism (SNP) *rs2802292*, suggesting the existence of an HSF1-FOXO3a axis involved in stress response pathways in human cells [89]. 

The mechanism of HSF1-FOXO3a interaction will be discussed in the next section of this review. HSF1 activity is regulated not only by its binding to HSEs but also by PTMs (such as phosphorylation by p38 MAPK), indicating the involvement of multiple factors in HSF1 modulation, besides HSPs [90]. 

Several findings suggest that FOXO3a takes part in the heat shock response pathway and in HSF1-mediated response to stress. In fact, it has been demonstrated that upon oxidative stress, FOXO3a and its invertebrate orthologs regulate and are regulated by a wide array of HSPs. For example, in *C. elegans*, it was shown that DAF-16 acts upstream of several HSPs in promoting lifespan extension [91] and that it binds the same regulatory regions of some HSP genes covered by HSF1, activating their expression in response to heat shock [92]. In *D. melanogaster*, it has been demonstrated that dFoxO can directly regulate the expression of a wide range of small and large HSPs, including the stress-inducible family of HSP70-related chaperones [15]. Studies performed in mammalian systems confirmed the involvement of FOXO3a in the heat shock response. For example, in mice, heat stress directly up-regulates Foxo3a through the activation of Hsp72 [93]. In human cultured cells, HSP70 regulation by FOXO3a has been shown to play a role in cell viability after exposure to different stresses. Indeed, up-regulation of HSP70 by FOXO3a has been found to promote neuronal cell death after ischemic injury [94]; conversely, down-regulation of HSP70 by FOXO3a in endothelial cells induces apoptosis after heat shock [95].

Other evidence indicates that upon heat stress, a significant increase occurs in FOXO3a phosphorylation (Ser 253) by the PTEN/AKT and MEK/ERK pathways, a signal that may promote FOXO3a proteasomal degradation [93].

### 2.5. Upstream Regulation of FOXO3a in Genotoxic Stress

Genome integrity is constantly challenged by endogenous DNA-damaging stresses, such as oxidative stress, external chemical and physical agents, ionizing radiation, heat shock, and osmotic stress. If left unrepaired, DNA lesions can be harmful, leading to permanent and inheritable alterations, which range from single point mutations to loss of chromosomic regions, in the DNA sequence. Cells have evolved fine and efficient molecular mechanisms to control and ensure DNA integrity. This control is achieved thanks to multiple and partially overlapping signaling pathways, such as DNA damage signaling, cell cycle checkpoints, DNA repair, and apoptosis. Detection of DNA damage triggers the induction of a signal transduction cascade, which ultimately arrests or slows down the cell cycle in order to create a time window for repair to be completed prior to DNA replication or cell division [96].

DNA damage can influence the activity of FOXO3a in different ways. Importantly, FOXO3a was found to modulate the expression of several genes that regulate cellular stress response at the G2-M cell cycle checkpoint. In particular, FOXO3a induces the expression of growth arrest and DNA damage 45 (GADD45) in response to stress stimuli that induce DNA damage [26]. Moreover, FOXO3a has been shown to down-regulate forkhead box M1 (FOXM1) through transcriptional and post-transcriptional mechanisms in response to genotoxic stress, thereby triggering cell cycle checkpoints at the G1/S, S, G2/M, and M phases upon DNA damage [97].

In addition to its well-known transcriptional regulation of genes involved in the response to stress stimuli, FOXO3a has also been shown to be part of functional networks that sense DNA damage. Specifically, FOXO3a directly interacts with ataxia telangiectasia mutated protein kinase (ATM), a kinase that orchestrates the activation and recruitment of downstream DNA damage response (DDR) factors to induce cell cycle arrest and repair [98]. 

Another PTM that regulates FOXO3a functions in response to genotoxic stress conditions is methylation. It has been reported that the methyl-transferase SET domain protein 9 (SET9) directly methylates FOXO3a in vitro and in cellulo on lysine 271, which is also a target of deacetylation by sirtuin 1 (SIRT1) [99]. FOXO3a methylation by SET9 decreases its protein stability but also moderately increases its transcriptional activity [99]. Modulation of FOXO3a stability and activity by methylation may be critical for fine-tuning cellular responses to genotoxic stress stimuli, thereby affecting FOXO3a ability to promote tumor suppression and longevity.

All these findings indicate that in mammals, FOXO3a regulates cell resistance to stress by inducing DNA repair and thereby influencing an organism’s lifespan. 

## 3. Architecture of FOXO3a Domains and its Nuclear Functions

Human *FOXO3* comprises three introns and four exons, of which only exon 2 and 3 code for the full-length 673-amino acid protein, which contains an N-terminal winged-helix/forkhead (FH) domain that is conserved among all FOX proteins. The FH domain of FOXO3a consists of three α-helices (H1–H3), three β-strands (S1, S2, S3), and two large unstructured wings (W1, W2) [98,100]. Crystal structure analysis revealed that the H3 winged-helix region of the FH is the primary element for DNA recognition and is capable of binding the consensus DNA sequence 5′-TTGTTTAC-3′, which is termed the forkhead response element (FRE) [101]. FOXO3a also contains three additional conserved regions, designated as CR1–CR3. The CR3 domain is an acidic region found at the C-terminus and has been shown to function as a transactivation domain that is required for coactivator recruitment [102,103]. FOXO3a is highly expressed in various tissues, including brain, heart, kidney, and spleen [104]. It controls the expression of a complex network of genes and can act either as a transcriptional activator or as a repressor, probably recruiting a wide range of cofactors during DNA binding [105]. Genomic studies indicated that transcriptional regulation by FOXO3a may be determined by specific epigenetic programs that are active at the moment of FOXO3a induction. Indeed, FOXO3a has been found to bind to open chromatin at enhancer regions, promoting enhancer activation [106]. In addition, FOXO3a-mediated enhancer activation correlates with the regulation of adjacent genes and the pre-existence of chromatin loops between FOXO3a-bound enhancers and target genes [106,107]. These findings highlight the importance of chromatin architecture in FOXO3a transactivation activity to orchestrate differential expression programs modulating a variety of cellular functions, including cell death, proliferation, metabolism, and ROS scavenging. Epigenetic alterations caused by environmental changes or genetic variation within regulatory regions are thought to be involved in the pathogenesis of and predisposition to aging and disease. Several genetic association studies have identified SNPs that confer protection against age-related diseases in humans, corroborating the evidence that *FOXO3* is a longevity-associated gene [23]. Importantly, previous reports failed to find an association between *FOXO3*-coding variants and longevity [108]. Conversely, among non-coding genetic variants, the G-allele of the rs2802292 SNP, located in *FOXO3* intron 2, has been reported to be strongly associated with longevity in different human populations [109,110]. A recent work showed that the 90-bp sequence around rs2802292 has an enhancer function. Indeed, the rs2802292 G-allele creates a unique HSE DNA-binding site for HSF1, inducing the expression of FOXO3a in response to different types of cellular stress (nutrient, genotoxic, and oxidative stress) [89]. These findings indicate the existence of an HSF1-FOXO3a axis that could be involved in stress response pathways in human cells, functionally regulating lifespan and disease susceptibility [89]. 

A recent study suggested that the G-allele of rs2802292 is part of a longevity haplotype that comprises nine other *FOXO3* intronic SNPs. The authors proposed a mechanism by which the pro-longevity haplotype favors gene–gene interaction between *FOXO3* and 46 flanking genes involved in various processes that contribute to cell resilience (autophagy, stress response, energy/nutrient sensing, cell proliferation, apoptosis, and stem cell maintenance). It was thus hypothesized that the genes implicated in this chromatin hub work coordinately to maintain cellular homeostasis and consequently healthy aging [111]. 

Altogether, these studies indicate that genetic variation in *FOXO3* regulatory regions may prove to be a major factor in the modulation of FOXO3a function and transcriptional output.

## 4. Roles of Nuclear FOXO3a in Cellular Stress Response

### 4.1. Role of FOXO3a in the Regulation of Genes Involved in Metabolic/Energetic Stress

Several reports have shown that FOXO3a is involved in the regulation of a number of insulin-responsive genes. Insulin affects the expression of a wide range of genes, many of which are regulated by this hormone at the transcriptional level. A consensus insulin response element (IRE) with a T(G/A)T-TT(T/G)(G/T) core sequence is associated with insulin-induced transcriptional repression of a variety of metabolic genes, including those encoding for phosphoenolpyruvate carboxykinase (PCK1), insulin growth factor-binding protein (IGFBP-1), tyrosine aminotransferase (TAT), glucose-6-phosphatase (G6PC), apolipoprotein C III (APOC3), and aspartate aminotransferase (GOT). Overexpression of FOXO3a promotes its binding to the IREs of PCK1 and IGFB-1, activating their transcription. This transactivation is inhibited by insulin via an AKT/PKB-dependent pathway [112]. Another study showed that FOXO3a binds to the IRE of the *G6PC* gene and stimulates its promoter activity in vitro [113]. Moreover, in an elegant study conducted on *Foxo1* and *Foxo3* liver-specific knock-out mice, deletion of both genes was shown to have synergistic effects on lowering the expression of G6PC, which is located in the endoplasmic reticulum and catalyzes the terminal reaction of both glycogenolysis and gluconeogenesis, (e.g., the conversion of glucose 6-phosphate into glucose). In addition, inactivation of Foxo3 was found to promote the expression of genes involved in lipogenesis, including F-box synaptic protein (*Fsn*) and 3-hydroxy-3-methylglutaryl-CoA reductase (*Hmgcr*), which encode two key enzymes regulating the synthesis of triglycerides and cholesterol. Overall, this study provided evidence that Foxo3 and Foxo1 inactivation serves as a potential mechanism by which insulin reduces hepatic glucose production and increases hepatic lipid synthesis and secretion in healthy and diabetic states [114].

The insulin-like growth factor 1 (IGF1)-AKT pathway controls muscle growth via mTOR and FOXOs. FOXO transcription factors play a critical role in muscle atrophy processes, being involved in the expression of rate-limiting enzymes of the ubiquitin-proteasome and autophagy-lysosome systems. Therefore, a fine regulation of FOXOs is critical to avoid excessive proteolysis and cachexia. 

Importantly, acetylation of FOXO3a by the histone acetyl-transferase p300 negatively regulates FOXO3a-mediated up-regulation of the F-box protein 32 (*FBXO32*)/*Atrogin-1* gene, which encodes a subunit of the SCF (SKP1/CUL1/F-box protein) ubiquitin-ligase, a complex involved in the degradation of muscle cell proteins during muscle atrophy [115]. Importantly, FOXO3a regulates the expression of ATP-binding cassette sub-family A member gene 6 (*ABCA6*), which encodes a cholesterol-responsive putative transporter protein that is thought to be involved in lipid homeostasis. ABCA6 mRNA is suppressed by IGF1, which stimulates phosphorylation and inactivation of FOXOs, while inhibition of PI3K has the opposite effect [116]. In low glucose conditions, FOXO3a directly activates tripartite motif containing 63 (*TRIM63*) gene transcription, acting synergistically with SMAD3 [117,118]. TRIM63 is an E3 ubiquitin ligase involved in muscle protein degradation and muscle atrophy during starvation, similar to Atrogin-1 [119].

### 4.2. Role of FOXO3a in the Regulation of Genes Involved in Oxidative Stress

FOXO3a has been reported to protect cells from oxidative stress by increasing ROS scavengers, thereby contributing to extending organismal lifespan. Kops and colleagues showed that FOXO3a, via PKB regulation, protects quiescent cells from oxidative stress, increasing the expression of MnSOD (manganese superoxide dismutase), a mitochondrial antioxidant enzyme converting superoxide, which is formed as a by-product during ATP generation in mitochondria, into hydrogen peroxide [120]. FOXO3a also mediates the regulation of CAT (catalase), another antioxidant enzyme catalyzing the conversion of hydrogen peroxide into water and oxygen [121]. Thus, FOXO3a is able to coordinately regulate the expression of MnSOD and CAT to decrease oxidative damage and increase cellular survival. Besides regulating ROS scavengers, FOXO3a can decrease ROS production through the suppression of mitochondrial function. Intriguingly, FOXO3a inhibits mitochondrial gene expression by inducing the expression of MAX dimerization proteins (MAD/MXD) and by modulating c-MYC protein stability. Consistently, FOXO3a activation has been shown to reduce the mtDNA copy number, decrease expression of mitochondrial proteins and lower the levels of respiratory complexes, reducing ROS production [67]. Furthermore, FOXO3a can protect vascular endothelial cells from oxidative stress through direct interaction with PGC-1α (PPAR-γ co-activator 1α), a well-characterized positive regulator of mitochondrial function and oxidative metabolism. PGC-1α regulates the transcription of a group of genes involved in ROS detoxification. 

Co-immunoprecipitation and in vitro interaction assays showed that FOXO3a and PGC-1α interact directly to bind the same promoter regions and induce the expression of stress protection genes. Moreover, the expression of PGC-1α is directly regulated by FOXO3a, suggesting that an auto-regulatory loop regulates FOXO3a/PGC-1α control of oxidative stress response [122]. 

Notably, oxidative stress can also trigger components of the heat shock response. Consistently, in humans, FOXO3a has been shown to play a role in the protection of neuronal tissue from ischemic stroke through modulation of HSP70 [94].

Finally, a recent study demonstrated the existence of an HSF1-FOXO3a-MnSOD/CAT/GADD45A cascade involved in cellular stress response and survival by promoting ROS detoxification, redox balance, and DNA repair [89].

### 4.3. Roles of FOXO3a in the Regulation of Hypoxic Stress Response

FOXO3a transcriptional activity plays an important role in cellular response to hypoxia. Activation of FOXO3a after hypoxia can promote cell death by up- and down-regulating a set of nuclear-encoded mitochondrial genes. In particular, it has been proposed that FOXO3a directly antagonizes c-MYC at promoters of nuclear-encoded mitochondrial genes (like aconitase 2, ACO2; ATP synthase membrane subunit C, ATP5G1; electron transfer flavoprotein dehydrogenase, ETFDH; mitochondrial ribosomal protein 12, MRPL12 and leucyl-tRNA synthetase 2, LARS2), thereby supporting cell adaptation to hypoxia through the regulation of mitochondrial structure and activity [78]. In fact, although c-MYC is a proto-oncogene that drives cell proliferation, it is also known to sensitize cells to cell death in hypoxic conditions [123]. At the same time, FOXO3a can also induce apoptosis through the up- and down-regulation of pro-apoptotic (BIM and Bcl2 associated X protein, BAX) and anti-apoptotic (B-cell lymphoma 2, BCL-2 and Bcl extra large) proteins, respectively [124]. 

In addition, FOXO3a regulates cell survival by inhibiting HIF1 transcriptional activity through an intricate feedback loop mechanism. In particular, it has been demonstrated that, under hypoxic conditions, FOXO3a can form a complex with HIF and p300, resulting in the suppression of HIF1 target genes, many of which promote apoptosis (*BNIP3*, Phorbol-12-myristate-13-acetate-induced protein 1, *PMAIP1*/*NOXA*, and DNA damage-inducible transcript 4 protein, DDIT4/*RTP801*) [79,81]. In parallel, FOXO3a can also inhibit HIF activity in more indirect ways, for example, by promoting the inactivation of ERK signaling [125] or by inducing the p300 cofactor CBP/p300 interacting transactivator 2 (CITED2) [79]. 

### 4.4. Roles of FOXO3a in the Regulation of Genotoxic Stress

FOXO3a plays a key role in the control of DNA damage through the regulation of different genes involved in sensing DNA damage, propagating response signals, and mediating cell cycle checkpoint arrest and DNA repair. Consistently, mammalian FOXO3a has been shown to trigger the G2/M checkpoint and DNA repair after oxidative stress. In particular, cell cycle arrest and DNA repair are directly regulated by FOXO3a via a GADD45a- and ATM-dependent mechanism [26]. Further studies indicate that FOXO3a may modulate DDR through direct binding to the ATM kinase, activating its signaling function [98]. FOXO3a also induces ATM expression in hemopoietic stem cells, further supporting a functional connection between the two factors [126].

In addition, FOXO3a has been shown to play a role in DNA damage-induced apoptosis through its interaction with the ATM-checkpoint kinase 2 (CHK2)-p53 pathway [98,126]. FOXO3a interaction with components of the ATM-CHK2-p53 pathway enhances the phosphorylation of this complex and induces the formation of nuclear foci at damaged sites, suggesting an important connection between FOXO3a and the ATM-CHK2-p53-mediated apoptotic program following DNA damage [66].

FOXO3a has been shown to negatively regulate the expression of FOXM1, a forkhead protein involved in the regulation of genes acting in DDR, and to compete for the binding of the same DNA motifs in target promoters (e.g., vascular endothelial growth factor, VEGF), thereby repressing the transcription of FOXM1 target genes. In particular, FOXO3a activation triggers the G1/S and G2/M cell cycle checkpoints, as well as repressing DDR, cell proliferation, and survival [97].

FOXO3a is also part of a signal transduction crosstalk with p53. Activation of FOXO3a by serum starvation induces its physical interaction with p53, decreasing p53 DNA binding activity and promoting its cytoplasmic translocation. In the cytoplasmic compartment, p53 is directed to the mitochondrial outer membrane, where it interacts with BCL-2 and BCL-XL, inducing the mitochondria-associated apoptotic pathway. Overall, FOXO3a together with p53 creates an effective tumor suppressor network that protects cells from DNA damage, playing an important role in genomic stability and prevention of tumor initiation [127]. 

### 4.5. FOXO3a: A Regulator of Cell Fate in Stress Conditions 

Two important cellular processes, such as cell cycle arrest and cell death, are regulated by FOXO3a through the induction of cell cycle suppressor gene transcription. Several studies have shown that cytokine or growth factor deprivation promotes the activation of FOXO3a, resulting in up-regulation of multiple cell cycle suppressor genes implicated in the induction of cell cycle arrest at the G1/S boundary. Conversely, in response to growth factors, FOXO3a is phosphorylated by AKT; this triggers FOXO3a nuclear export and cytoplasmic sequestration, thereby interfering with its nuclear transcriptional functions. Activation of FOXO3a was found to promote increased transcription of the *p27* gene, resulting in inhibition of proliferation [24]. A study conducted in murine WEHI 231 immature B lymphoma cells showed that suppression of the Pi3k signaling pathway leads to induction of Foxo3a and inhibition of c-Myc, which coordinately and inversely control the regulation of *p27* gene transcription [128]. Another FOXO3a target is the cyclin dependent kinase inhibitor p21 (p21), a member of the CDK interacting protein/kinase inhibitory protein (Cip/Kip) family. In response to transforming growth factor β (TGFβ), the transcription factors mothers against decapentaplegic homolog 3 (SMAD3) and mothers against decapentaplegic homolog 4 (SMAD4) physically interact with the DNA binding domain (DBD) of FOXO3a, forming a complex that induces p21-mediated cell cycle arrest (at the G1/S transition) of epithelial, neuronal, and immune cells [25,129]. In addition to p21 and p27, FOXO3a also up-regulates p15 and p19, which are part of the INK4 family of cyclin-dependent kinase (CDK) inhibitors, via a PI3K/AKT-dependent axis. Importantly, FOXO3a transcriptionally regulates the expression of several target genes, such as *BIM*, *NOXA*, TNF-related apoptosis-inducing ligand (*TRAIL*), p53 upregulated modulator of apoptosis (*PUMA*), and Fas ligand (*FASL*), that promote apoptosis. In cultured neuroblastoma cells, activation of FOXO3a induces its binding to the promoter region of *BIM* and *NOXA*, causing sympathetic neuron cell death [130]. Moreover, FOXO3a directly regulates the extrinsic apoptotic pathway by enhancing the transcription of apoptotic factors, such as FASL and TRAIL in cerebellar neurons, leukemic, and prostate cancer cells [27,131,132]. FOXO3a is also associated with the regulation of the proapoptotic protein PUMA in lymphoid cells [133]. 

Overall, these findings demonstrate that FOXO3a-dependent cell cycle arrest and apoptosis induction are important for the regulation of cell proliferation and survival and further suggest that FOXO3a pathological alteration can contribute to uncontrolled cell proliferation and to the acquisition of an apoptosis-resistant cell phenotype.

## 5. Role of Mitochondrial FOXO3a in Cellular Stress Response

Mitochondria are crucial organelles for metabolic, oxidative, and genotoxic homeostasis in eukaryotic cells. 

The first evidence of extra-nuclear localization of a FOXO family member was reported in *C. elegans* a few years ago in a study in which the authors described an AMPK-dependent dietary restriction pathway that did not stimulate nuclear accumulation of the FOXO homolog DAF-16, suggesting a functional cytoplasmic role for this transcription factor [134].

Subsequently, FOXO3a was described to co-precipitate with the mitochondrial sirtuin SIRT3 in the mitochondrial fraction of mammalian cells [135]. SIRT3 is reported to be localized almost exclusively in the mitochondria, where it acts as the primary deacetylase [136]. SIRT3 deacetylase activity does not appear to be required for FOXO3a mitochondrial localization but is needed for the regulation of FOXO3a DNA binding ability and activation of the mitochondrial pathway in oxidative stress response [137]. Further studies confirmed the occurrence of phosphorylation-dependent FOXO3a mitochondrial shuttling in response to metabolic stress stimuli [138]. Notably, a phosphorylation code that regulates FOXO protein intracellular trafficking and is evolutionarily conserved from invertebrates to humans has been identified [31]. 

Mitochondrial FOXO3a (mtFOXO3a) was identified as a shorter FOXO3a isoform that seems to be processed similarly to canonical mitochondrial proteins [47]. mtFOXO3a loses residues 1–148 of the N-terminal domain but retains an intact DBD (amino acids 149–242), thus being able to bind to mtDNA and activate the expression of mitochondrial genes [47]. In particular, FOXO3a N-terminus (aa 1–148) was shown to be required for proper recruitment to the mitochondria, with residues 98–108 being strictly necessary for FOXO3a cleavage and import into the mitochondrial matrix. Interestingly, this region is specific to FOXO3a, since it is not conserved in other human FOXO members but is evolutionarily conserved across species [47]. 

### 5.1. Mitochondrial FOXO3a in Metabolic Stress Response

FOXO3a is a key player in the transduction of mitochondrial signals in response to nutrient shortage, especially under GR conditions. A high adenylate energy charge (AMP/ATP ratio) value is the major indicator of metabolic stress and is responsible for the activation of cellular energy sensor pathways, which are primarily mediated by AMPK. In eukaryotes, the energy sensor AMPK is activated by metabolic stress and restores intracellular ATP levels by switching off glucose and lipid anabolic processes (such as gluconeogenesis, glycogen synthesis, and fatty acid biosynthesis) and switching on the corresponding catabolic pathways (aerobic glycolysis, glycogenolysis, and fatty acid β-oxidation) [139]. 

Recent findings revealed an unprecedented mechanism that supports mitochondrial energy production during nutrient restriction and links the mitochondrial AMPK-FOXO3a-SIRT3 pathway to the beneficial effects of limiting nutrient intake in mammals [47,138]. In particular, accumulation of mtFOXO3a was described under GR conditions in various established and primary muscle and fibroblast cell lines of murine and human origin [138].

Upon metabolic stress (GR), AMPK directly modulates FOXO3a functions by phosphorylating six specific FOXO3a residues [38], and growing experimental evidence suggests that other sites may also be involved [140].

In the same conditions, AMPK activates SIRT1, indirectly modulating FOXO3a transcriptional activity [141,142]. 

Overall, these data suggest a correlation between AMPK, FOXO3a, and sirtuins in the mitochondrial response to metabolic stress and support the existence of a mitochondrial arm of the AMPK-FOXO3a axis [138]. Subsequent findings confirmed that functional AMPK is required for proper GR-dependent mtFOXO3a accumulation. Indeed, in normal cells and tissues subjected to metabolic stress, FOXO3a Ser 30 phosphorylation by AMPK was required to promote FOXO3a translocation into the mitochondria [47]. 

Interestingly, inside the mitochondria, FOXO3a interacts with mtRNA polymerase and SIRT3 [47,138]. SIRT3-FOXO3a mitochondrial interaction is required for the formation of a FOXO3a-SIRT3-mtRNA polymerase complex bound to mtDNA and is necessary to activate the transcription of mitochondrial-encoded catalytic subunits of the respiratory chain complexes and oxidative phosphorylation system [135,138].

This mitochondrial arm of the AMPK-FOXO3a axis functions as a recovery mechanism to sustain cellular metabolic homeostasis upon nutrient shortage in normal cells. In this way, normal cells react to metabolic stress by restoring the biochemical balance between glucose anabolic/catabolic processes and the AMP/ATP ratio [138,143].

Recently, the molecular mechanisms that allow FOXO3a recruitment to the mitochondrial surface and its subsequent import into the organelles of cancer cells and tumors subjected to GR have been characterized. FOXO3a mitochondrial import occurs in a MEK/ERK- and AMPK-dependent manner, and the N-terminal domain of FOXO3a (amino acids 1–148) is essential in this mechanism [47]. Of note, FOXO3a N-terminus residues 98–108 contain overlapping consensus motifs for mitochondrial processing peptidase (MPP) and mitochondrial intermediate peptidase (MIP) [144] and are required for FOXO3a cleavage and subsequent mitochondrial import. Upon metabolic stress, mtFOXO3a is also required to sustain the membrane potential of functional and healthy mitochondria in cancer cells [47]. 

In response to GR, activated MEK/ERK and AMPK pathways phosphorylate FOXO3a serine residues Ser 12 and Ser 30, respectively. Interestingly, FoxO3A accumulation into the mitochondria requires both Ser 12 and Ser 30 phosphorylation in metabolically stressed tumor cells, while it only needs AMPK phosphorylation at Ser 30 in normal cells [47]. Several studies of massive proteomic sequencing in human cancer samples confirmed phosphorylation at both these FOXO3a serine residues [145,146,147,148,149,150]. These phosphorylation marks are the triggering signals that lead to FOXO3a translocation into the mitochondrial matrix, where it binds to mtDNA together with TFAM, mtRNA polymerase, and SIRT3, and activates the expression of mitochondrial oxidative phosphorylation (*OXPHOS*) genes involved in sustaining and re-establishing the normal energetic state of mitochondria in metabolically stressed cancer cells. Notably, combined treatment with a MEK inhibitor (Trametinib, which is approved by the Food and Drug Administration, FDA, for clinical use) and an AMPK inhibitor (Compound C) triggers a synergistic cytotoxic effect and potentiates the antitumor activity of each single agent in metabolically stressed cancer cells [47]. Interestingly, in these cells, the substitution of FOXO3a serines (Ser 12, Ser 30) with alanines—which cannot be phosphorylated—prevents FOXO3a mitochondrial import, contributing to BIM transcription-mediated apoptosis induction [47].

The crosstalk between the MEK/ERK and AMPK cascades converging on the FOXO3a N-terminal domain represents the first evidence of an emerging mitochondrial FOXO code. 

### 5.2. Mitochondrial FOXO3a-Mediated Chemoresistance in Chemotherapy-Dependent Genotoxic Stress

Several studies highlighted the nuclear role of FOXO3a in genotoxic stress resistance, but recent findings suggest that mtFOXO3a may also play a key role in cellular response to genotoxic stress.

A recent study carried out in *C. elegans* showed that in younger worms, transcription factor cep 1, CEP-1 promotes nuclear DAF-16 activation to induce protective stress response to DNA damage, while CEP-1/p53-dependent loss of DAF-16 nuclear retention was observed upon chronic genotoxic stress in older worms. This change in DAF-16 subcellular localization suggests a potential extra-nuclear role for DAF-16 in chronic genotoxic stress resistance [151].

Of note, in the first study describing the mitochondrial localization of DAF-16 homolog FOXO3a, its activation was found to be mediated by SIRT3 acetylation upon oxidative and genotoxic stress [134,135].

Overall, this evidence suggests a potential extra-nuclear role for FOXO3a in genotoxic stress resistance, probably mediated by SIRT3.

Surprisingly, Celestini et al. showed that mtFOXO3a could represent a survival factor in cancer cells subjected to metabolic and genotoxic stress induced by chemotherapeutic treatments [47]. Indeed, similar to the mtFOXO3a pro-survival activity described in cancer cells under metabolic stress, a mtFOXO3a-dependent chemoresistance mechanism was observed in cancer cells and tumors treated with anti-neoplastic agents [47] that affect DNA/RNA structural and functional integrity, such as irinotecan (CPT-11) [152], cisplatin (CDDP) [153], 5-fluorouracil (5-FU) [154], and etoposide (VP-13) [155]. Besides, in these conditions, mitochondrial accumulation of FOXO3a was found to be dependent on the activation of the MEK/ERK but not the AMPK pathway. This finding was also confirmed in xenografted tumors treated with CDDP [47]. Conversely, mtFOXO3a was found to be required for apoptosis induction by metformin through an AMPK-dependent mechanism [47].

The chemoresistance functions of mtFOXO3a in response to chemotherapy will need further investigation. Indeed, a thorough understanding of the underlying mechanisms might help devise molecularly targeted therapeutic strategies aimed at manipulating FOXO3a functions in cellular metabolism to counteract cancer initiation and progression.

## 6. FOXO3a-Dependent Nuclear and Mitochondrial Crosstalk in Cellular Stress Response

Cellular stress response is coordinated through the communication between mitochondria and the nucleus. In this regard, the dual subcellular localization of FOXO3a seems to support an accurate and efficient response to stress. In particular, FOXO3a finely orchestrates differential nuclear/mitochondrial expression programs regulating a variety of cellular functions, such as cell death, proliferation, metabolism, and ROS scavenging, in response to several stressors. Notably, FOXO3a nuclear and mitochondrial target genes may have synergistic or antagonistic effects in normal and/or stressed cells, suggesting that FOXO3a is an adaptable player in the dynamic processes that regulate homeostasis.

### 6.1. Nuclear FOXO3a-Dependent Effects in Stressed Mitochondria

Nuclear FOXO3a has various direct and indirect effects on mitochondrial metabolism under stress conditions. Indeed, FOXO3a is involved in molecular processes controlled by the nucleus to regulate mitochondrial function through anterograde signals in response to various stress stimuli. These anterograde signals are controlled by upstream sensors that detect changes in stress and metabolic conditions.

External stressors, such as GR, oxidative imbalance, and hypoxia, induce an increase in AMP and NAD+ levels in the intracellular environment of stressed cells. The AMP/ATP and NAD+/NADH ratios are the energetic and redox sensors, respectively, of metabolic and oxidative stress signaling [156,157]. High levels of AMP and NAD+ trigger mitochondrial respiration through stress response pathways mediated by the AMPK-FOXO3a and MAPK-FOXO3a axes. Cellular response to metabolic stress is finely regulated by the balance between the AMPK and PI3K/AKT pathways, which exert antagonistic effects on FOXO3a activity. In particular, under GR conditions, high levels of AMP promote the AMPK-dependent phosphorylation of FOXO3a, activating the nuclear transcription of energetic stress-responsive FOXO3a target genes that promote mitochondrial activity in order to restore normal glucose and ATP levels. In liver cells, FOXO3a promotes transcription of the gluconeogenic genes *G6PC* and *PCK1*, which enable steady glucose blood levels during fasting [112,113]. FOXO3a-mediated induction of *G6PC* and *PCK1* genes is negatively regulated by insulin-induced PI3K/AKT signaling [158].

Under metabolic stress, activation of AMPK increases NAD+ levels, which consequently induces the activation of SIRT1 [141]. Activation of SIRT1, a deacetylase that regulates metabolic activity in response to cellular stress, promotes the deacetylation of its downstream targets, including PGC-1α, FOXO1, and FOXO3a. Overall, these regulating mechanisms lead to the activation of mitochondrial energy metabolism. In aged mouse kidneys, calorie restriction (CR) induces SIRT1-dependent mitophagy and attenuates hypoxia-associated mitochondrial and renal damage [159]. Further analyses showed that Sirt1 deacetylates Foxo3a, and Foxo3a induces mitophagy by promoting expression of Bnip3 [159]. Bnip3 is an atypical BH3-only protein known as a pro-apoptotic factor that induces mitochondrial dysfunction, but surprisingly, it can also protect cells by inducing mitophagy [160].

In addition to regulating mitophagy, FOXO3a can inhibit mitochondrial fission and apoptosis by transactivating mir-484, a microRNA that targets the mitochondrial fission protein Fis1 [161].

It has been shown that under GR conditions, AMPK promotes FOXO3a accumulation into the mitochondria of fibroblasts and skeletal myotubes, inducing the formation of a protein complex containing mtFOXO3a, SIRT3, TFAM, and mtRNA polymerase at mitochondrial DNA-regulatory regions, promoting the activation of mitochondrial gene expression and a subsequent increase in mitochondrial respiration [47,138]. Of note, AMPK can be triggered by mitochondrial ROS through an alternative pathway that involves the FOXO3a target genes *MnSOD*, *CAT*, and *PGC-1α* [122,140,162].

Mitochondria represent the central hubs for bioenergetics and oxidative catabolism and are an important source of endogenous ROS that are produced by the respiratory complexes. Overproduction of mitochondrial ROS can cause oxidative damage to mitochondrial DNA, proteins, and lipids, and in severe cases can lead to mitophagy through an SIRT1-dependent mechanism [163]. Moreover, mitochondrial ROS are involved in signaling pathways from the mitochondria to the nucleus and vice versa that are also mediated by FOXO3a. In particular, FOXO3a and other FOXO members play a crucial role in preventing oxidative damage to cell membranes and especially to internal and external mitochondrial membranes, which are particularly exposed to oxidative damage due to the presence of O_2_, the last electron and proton acceptor in the respiratory chain. Interestingly, several in vitro and in vivo studies suggested that the FOXO3 target MnSOD is a new type of tumor suppressor protein [164,165,166]. Moreover, in *C. elegans*, the transcriptional activation of MnSOD mediated by DAF-16 represents a major mechanism of resistance to oxidative stress and is associated with longevity [120].

FOXO3a can also indirectly control mitochondrial ROS production through inhibition of mitochondrial gene expression by inducing MAX interactor 1 (*MXI1*), a member of the MAD/MXD family of transcriptional repressors, which acts as a c-Myc antagonist [167]. Consistently, in response to oxidative stress, FOXO3a inhibits mitochondrial activity by lowering the mtDNA copy number, reducing the expression of mitochondrial genes through the down-regulation of TFAM expression and decreasing ROS production through the down-regulation of respiratory complexes [67]. Furthermore, FOXO3a can protect cells from oxidative stress through direct interaction with PGC-1α on the same promoter regions and by inducing genes involved in ROS detoxification. In particular, PGC-1α is a master transcriptional regulator of several mitochondrial functions, including their biogenesis program [168,169].

In response to oxidative stress, FOXO3a is phosphorylated by several kinases that promote its nuclear accumulation and the expression of ROS scavengers. In particular, MST1, which can directly phosphorylate FOXO3a, is also an upstream kinase of the JNK and p38 MAPK pathways, whose expression induces pro-apoptotic morphological changes in stressed cells. MST1 promotes two events through JNK activation: It induces the activation of caspases, triggering caspase-8-dependent apoptosis via the mitochondrial pathway, and it promotes chromatin condensation [170]. The downstream effect of MAPK-dependent phosphorylation of FOXO3a is the disruption of FOXO3a-14-3-3 interaction, which promotes FOXO3a nuclear translocation and the transcription of its ROS scavenger targets [171].

FOXO3a activity in the context of the nuclear–mitochondrial crosstalk is also modulated by sirtuin-dependent deacetylation. Intriguingly, the seven mammalian sirtuins (SIRT1-7) show different subcellular localization, suggesting an inter-organellar coordinated activity under different stress conditions. In particular, SIRT1, SIRT6, and SIRT7 are found predominantly in the nucleus, while SIRT3, SIRT4, and SIRT5 are primarily located in the mitochondria and SIRT2 in the cytoplasm [172]. In response to various stress stimuli, FOXO3a can be specifically deacetylated by SIRT1, SIRT3, and SIRT6, which are NAD+-dependent deacetylases [163].

As mentioned above, AMPK activates SIRT1 through the regulation of NAD+ levels. SIRT1 activates PGC-1α, which in turn induces the expression of mitochondrial genes through the interaction with peroxisome proliferator-activated receptors (PPARs), nuclear respiratory factors (NRFs), or estrogen-related receptors (ERRs) [173,174]. Besides, under oxidative and genotoxic stress, AMPK modulates FOXO3a transcriptional activity by activating SIRT1 [141]. 

FOXO3a deacetylation exerts a cell-dependent cytoprotective effect by inhibiting pro-apoptotic genes and concomitantly increasing the expression of GADD45α, MnSOD, and p27, which are downstream FOXO3a targets involved in cell cycle arrest, DNA repair, and antioxidant activity [142]. In humans, CBP/p300 can acetylate FOXO3a, thereby reducing its transcriptional activity. Of note, FOXO3a acetylation promotes apoptosis by enhancing the expression of pro-apoptotic FOXO3a target genes, such as *BIM*, *P21*, and fas ligand 6 (*FASL6*) [142].

SIRT6/FOXO3a interaction exerts a protective effect against ROS formation, which is a major mechanism of cardiac cell damage in ischemia/reperfusion (I/R) injury. In particular, mitochondrial SIRT6 up-regulates the AMP/ATP ratio and then activates the AMPK-FOXO3a axis. In this manner, SIRT6 promotes the FOXO3a-mediated expression of antioxidant-encoding genes (*MnSOD* and *CAT*) [175]. The increase in cellular ROS levels induces a transcriptional program encompassing FOXO3a target genes, including *BIM*, *BCL-XL*, and *survivin* [176]. In the mitochondria, the proteins encoded by these genes affect mitochondrial membrane potential, respiration, and cellular ROS levels. Unlike other anti-apoptotic proteins, such as BCL-XL, which do not alter mitochondrial functions, survivin leads to changes in mitochondrial architecture, respiration efficiency, and energetic metabolism. Indeed, it regulates mitochondrial fusion/fission machinery and mitochondrial respiration by interfering with complex I activity, and it reprograms the energetic metabolism by activating glycolysis to produce energy for survival [177]. Thus, FOXO3a both directly and indirectly affects mitochondrial respiration, ROS accumulation, and even mitochondrial shape, which on the one hand influences apoptosis sensitivity of tumor cells but may also have a significant impact on lifespan in multicellular organisms [176].

### 6.2. Mitochondrial FOXO3a-Dependent Effects in Stressed Cells

Current knowledge on FOXO3a mitochondrial activity and on its direct and indirect effects on cellular metabolism in dynamic cellular stress responses is far from being exhaustive; however, it suggests a potential role for FOXO3a in retrograde signaling from stressed mitochondria to the nucleus.

Under metabolic stress (GR conditions), high levels of AMP and NAD+ are primarily responsible for AMPK activation. Subsequently, specific AMPK-dependent phosphorylation of FOXO3a on Ser 30 promotes FOXO3a mitochondrial translocation [138]. In glucose-restricted normal and cancer cells, the synergistic activation of the MEK/ERK and AMPK pathways enhances mitochondrial respiratory activity through mtFOXO3a-mediated transcription of *OXPHOS* genes [47,138]. In these conditions, the expression of *OXPHOS* genes promotes mitochondrial ATP production to restore the AMP/ATP ratio and allows the accelerated consumption of NAD+ in an oxidative metabolism to restore the NAD+/NADH ratio [47]. Consequently, the activation of mitochondrial OXPHOS metabolism mediated by mtFOXO3a promotes the expression of catabolic enzymes in the nucleus.

SIRT3 is the only mitochondrial sirtuin able to deacetylate FOXO3a. It has been found to be a negative regulator of cardiac hypertrophy by activating FOXO3a-dependent antioxidants, i.e., MnSOD and CAT, as well as by suppressing ROS-mediated Ras activation and the downstream MAPK/ERK and PI3K/AKT signaling pathways [178]. The SIRT3-FOXO3a pathway has been widely reported to maintain mitochondrial homeostasis and cellular function. SIRT3 has been shown to activate autophagy by increasing the interaction of voltage dependent anion channel 1 (VDAC1) with Parkin [179], through AMPK signaling [180], or through the Sirt3/MnSOD pathway [181] via deacetylation of FOXO3a, with subsequent up-regulation of *BNIP3* and *BNIP3L* [182].

Moreover, the effect of FOXO3a and SIRT3 direct interaction depends on their subcellular localization. In the mitochondria, it promotes the transcription of oxidative stress response genes, such as *MnSOD* [120], *CAT* [183], peroxiredoxin III (*PRDX3*) [184,185], or the redox enzyme sestrin 3 (*SESN3*) [186,187]; conversely, in the nucleus, it enhances the transcription of FOXO3a nuclear target genes involved in metabolic and oxidative stress response (Figure 2) [188].

## 7. Conclusions

FOXO3a plays a key role in response to stress to maintain cellular homeostasis. This is accomplished through the coordinated regulation of its nuclear and mitochondrial functions. Indeed, different types of stressors can differentially modulate nuclear and/or mitochondrial FOXO3a activity, allowing cells to implement the most appropriate stress response.

To achieve this fine and adaptable regulatory function, FOXO3a activity is modulated by various mechanisms, which include several types of PTMs, feedback-loop systems, and shuttling between subcellular compartments. In particular, FOXO3a is kept inactive in the cytoplasm and relocates into the nucleus and/or the mitochondria upon activation, where it binds to and transactivates either nuclear or mitochondrial genes.

It is becoming more and more evident that both nuclear and mitochondrial FOXO3a functions are crucial for the maintenance of cellular homeostasis, not only because these organelles are equally important for cell survival but also because they can influence each other activities. In fact, while nuclear FOXO3a directly promotes mitochondrial adaptive pathways, mitochondrial FOXO3a can in turn control nuclear function through the induction of gene expression programs regulated by different metabolites. Thus, the role of FOXO3a in mediating nuclear–mitochondrial crosstalk adds a further level of complexity to the network of stress-activated pathways, ensuring a highly dynamic and adaptable cellular response to stress. On this basis, since alterations of cellular stress response pathways are a hallmark of many human pathological conditions, including aging-related diseases and cancer, it has been hypothesized that FOXO3a might represent a potential therapeutic target.

However, given the complexity of FOXO3a-mediated networks, the choice between its activation or inhibition as a therapeutic strategy is controversial, since both hyperactivation and impairment of specific stress response pathways can lead to detrimental outcomes. Thus, a better understanding is needed of the molecular mechanism underlying the different functions of FOXO3a in response to cellular stress. This would enable researchers to develop new strategies aimed at switching such responses from cell death into survival programs or vice versa, depending on the cellular context and pathological status.

## Figures and Tables

**Figure 1 cells-08-01110-f001:**
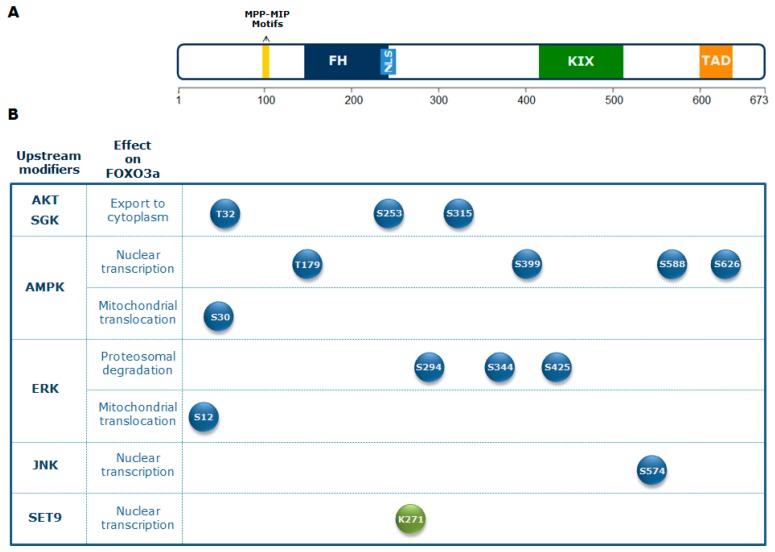
Post-translational regulation of FOXO3a. (**A**) Schematic representation of FOXO3a domains (MPP-MIP motifs, consensus motifs for mitochondrial processing peptidase (MPP) and mitochondrial intermediate peptidase (MIP); forkhead domain, FH; nuclear localization signal, NLS, Kinase-inducible domain interacting domain binding domain, KIX; transactivation domain TAD). (**B**) Summary of FOXO3a post-translational modifications (PTMs). Depicted are the most important upstream signals (AKT8 virus oncogene cellular homolog, AKT; serum and glucocorticoid-induced kinase, SGK;5′-AMP-activated protein kinase AMPK; c-Jun N-terminal kinase, JNK; extracellular signal-regulated kinase, ERK, SET domain protein 9, SET9) regulating FOXO3a subcellular localization and activity through reversible PTMs, which include phosphorylation (blue circle) and methylation (green circle) on specific amino acid residues (T, threonine; S, serine; K, methionine).

**Figure 2 cells-08-01110-f002:**
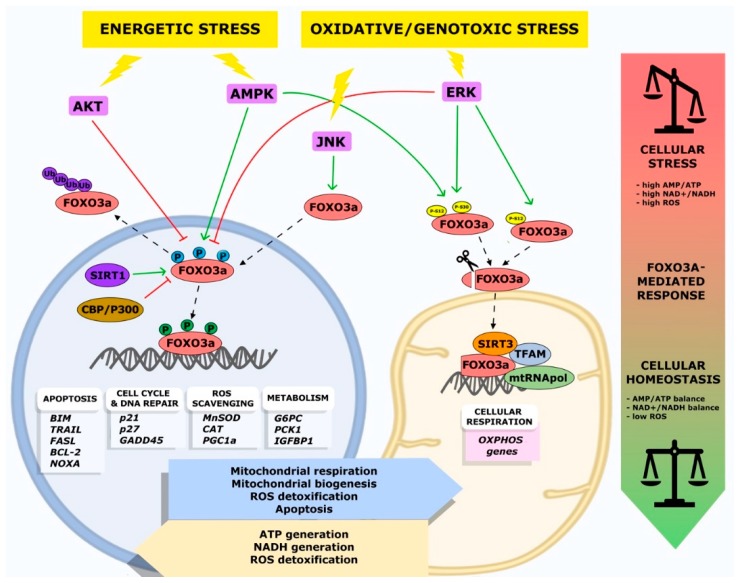
Schematic representation of Forkhead box O3 (FOXO3a)-mediated stress response. Perturbations of cellular homeostasis, such as nutrient shortage, high concentration of intracellular ROS, or genotoxic stress, activate FOXO3a upstream stress sensors (purple boxes), which in turn modulate FOXO3a subcellular localization and/or activity through various post-translational modifications (PTMs) (green arrows represent activation signals, red bar-headed lines represent inhibitory effects). In the cytoplasm, FOXO3a is inactive and is targeted for poly-ubiquitination, which leads to its further proteasomal degradation. Upon phosphorylation by c-Jun N-terminal kinase (JNK), FOXO3a is shuttled into the nucleus, where its transcriptional activity is further regulated by an activator (e.g., 5′-AMP-activated protein kinase, AMPK; sirtuin 1, SIRT1) or repressor (e.g., AKT8 virus oncogene cellular homolog AKT, CREB binding protein and p300 (CBP/p300) signals. Depending on the PTM pattern, FOXO3a orchestrates different transcriptional programs involved in several cellular processes, including apoptosis, cell cycle progression, DNA repair, reactive oxygen species (ROS) detoxification, and cellular metabolism. Recent evidence showed that metabolic stress or chemotherapy treatment can also promote AMPK- and extracellular signal-regulated kinase (ERK) dependent mitochondrial accumulation of a FOXO3a cleaved form, which activates the expression of mitochondrial oxidative phosphorylation (OXPHOS) genes involved in cell survival. The crosstalk between FOXO3a nuclear and mitochondrial functions is crucial for the restoration and maintenance of cellular homeostasis.

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
