# Peer review of "FOXO3a from the Nucleus to the Mitochondria: A Round Trip in Cellular Stress Response"

_cells, 2019, doi:10.3390/cells8091110_

Round 1

Reviewer 1 Report

In the reviewed manuscript, Fasano and colleagues present an elegant summary of the latest findings regarding the emergent role of FOXO3a transcription factor in the cellular stress response. Of particular interest is the compilation of the various functions of both nuclear and mitochondrial FOXO3a, as well as the nuclear-mitochondrial crosstalk.

The manuscript is well structured, clearly written, and addresses all the fundamental issues about FOXO3a proteins and its relationship with different types of stressors. The information provided is well supported with a large number of cited and commented articles.

The authors should however address and correct some points that would make the manuscript suitable for publication in Cells:

-On of the main concerns is the lack of figures, schemes or tables that could help the reader integrate all the revised information. One or more figures should be included depicting the relevant points/information of the various sections (upstream regulatory mechanisms, nuclear FOXO3a, mitochondrial FOXO3a…).

-The manuscript is very extensive and would benefit from shortening some of its parts. Information is sometimes unnecessarily repeated and this should be avoided (i.e. GADD45 as FOXO3a target explained in page 7 and in page 11, ATM-CHK2-p53 pathway in pages 8 and 11, MnSOD and CAT as FOXO3a targets in pages 5, 10 and 15…).

In this same line of thought, section 4 of the manuscript (Genetic variations and cell resilience) lies out of the focus of the manuscript. The authors should consider deleting this short section.

-As mentioned the manuscript contains a large group of references, some of which are not entirely appropriate and thus need to be revised and corrected.

 Page 3, lines 97-99. The processes regulated by FOXO3a should be supported with the original/seminal papers that described the main targets in the various processes: cell cycle progression (refs 130 and 133, not 24), DNA repair (ref 98, not 25), apoptosis (ref 37, not 28­)... References 25 to 28 are recent publications that do not describe the identification of main targets.

 Page 3, lines 128-130. The correct reference for this statement is ref 37 (seminal paper), not 34 (a recent review).

 Page 5, lines 216-218. The correct references are refs 123 and 124 (seminal papers), not 60 and 61.

Author Response

RESPONSE TO REVIEWER 1 COMMENTS

Point 1) On of the main concerns is the lack of figures, schemes or tables that could help the reader
integrate all the revised information. One or more figures should be included depicting the relevant
points/information of the various sections (upstream regulatory mechanisms, nuclear FOXO3a,
mitochondrial FOXO3a…).
We thank the reviewer for this useful suggestion. In this amended version, we integrated the revised
information with Figure 1, in which we reported the major PTMs targeting FOXO3a in response to
stress stimuli. Moreover, in accordance with the reviewer’s suggestion, we also included another
figure (Figure 2) that elucidate the relevant nuclear and mitochondrial mechanisms of stress response
mediated by FOXO3a in these subcellular compartments.
Point 2) The manuscript is very extensive and would benefit from shortening some of its parts.
Information is sometimes unnecessarily repeated and this should be avoided (i.e. GADD45 as
FOXO3a target explained in page 7 and in page 11, ATM-CHK2-p53 pathway in pages 8 and 11,
MnSOD and CAT as FOXO3a targets in pages 5, 10 and 15…).
In this same line of thought, section 4 of the manuscript (Genetic variations and cell resilience) lies
out of the focus of the manuscript. The authors should consider deleting this short section
We agree with the reviewer that some parts of our manuscript could be shortened. In this amended
version, we deleted the redundant parts indicated by the reviewer. In particular, we deleted redundant
information related to GADD45 (in page 11), the ATM-CHK2-p53 pathway (in page 8) and MnSOD
and CAT (in pages 6 and 15), while keeping only data that are essential to the logical flow of the
review.
The short section entitled “Genetic variations and cell resilience” describes the intriguing role of the
FOXO3 SNP rs2802292–HSF1 axis in response to different types of cellular stresses (nutrient,
genotoxic, and oxidative stress) and reports that the rs2802292 SNP takes part in a pro-longevity
haplotype. Considering the significant role of this SNP in cellular stress response and homeostasis,
we believe it is important to mention this mechanism. In this amended version of the manuscript, we
thus shortened and combined paragraphs 3 and 4 in a single section entitled “Architecture of FOXO3a
Domains and its Nuclear Functions”.
Point 3) As mentioned the manuscript contains a large group of references, some of which are not
entirely appropriate and thus need to be revised and corrected.
-Page 3, lines 97-99. The processes regulated by FOXO3a should be supported with the
original/seminal papers that described the main targets in the various processes: cell cycle progression
(refs 130 and 133, not 24), DNA repair (ref 98, not 25), apoptosis (ref 37, not 28-)... References 25
to 28 are recent publications that do not describe the identification of main targets.
-Page 3, lines 128-130. The correct reference for this statement is ref 37 (seminal paper), not 34 (a
recent review).
-Page 5, lines 216-218. The correct references are refs 123 and 124 (seminal papers), not 60 and 61.
We thank the reviewer for this useful recommendation. In this amended version of the manuscript,
we replaced the indicated references with the suggested seminal papers.

Reviewer 2 Report

This is a comprehensive compendium on the known roles of the transcription factor FOXO3 by Fasano and colleagues. This review addresses previously described roles of FOXO3 from C. elegans to humans and its targets modulating lifespan and stress responses under health and disease conditions. They cover the signal transduction axis involving ROS, PI3K/AKT and MAPKs in great detail.  

While the review is well-structured and balanced, this reviewer believes the authors should address a couple of concerns before this manuscript can be further considered.

1) The authors are missing the known roles of FOXO3 and circadian clocks. For example, please address the known roles of FOXO3 during autophagy (PMID: 22520961) and insulin-mediated signalling (PMID: 24856209 ).

2) Please consider adding a comprehensive diagram/figure comparing the roles of FOXO3 in the nucleus versus the mitochondrion. Perhaps a table would work as well. 

Author Response

RESPONSE TO REVIEWER 2 COMMENTS
Point 1) The authors are missing the known roles of FOXO3 and circadian clocks. For example, please address
the known roles of FOXO3 during autophagy (PMID: 22520961) and insulin-mediated signalling (PMID:
24856209 ).
We are grateful to the reviewer for pointing out this missing information. In this amended version of the
manuscript (page 4, lines 185-192), we described the role of FOXO3a in autophagy (PMID: 22520961) and
insulin-mediated signaling (PMID: 24856209) within circadian clock regulation.
Point 2) Please consider adding a comprehensive diagram/figure comparing the roles of FOXO3 in the nucleus
versus the mitochondrion. Perhaps a table would work as well.
We thank the reviewer for this useful suggestion. In this amended version, we integrated the revised
information with Figure 1, in which we reported the major PTMs targeting FOXO3a in response to stress
stimuli. Moreover, in accordance with the reviewer’s suggestion, we also included another figure (Figure 2)
that elucidate the relevant nuclear and mitochondrial mechanisms of stress response mediated by FOXO3a in
these subcellular compartments.